# Bicarbonate concentration as a predictor of prognosis in moderately severe COVID-19 patients: A multicenter retrospective study

Ken-ei Sada[1,2,3]☯*, Ryohei Yamamoto[4]☯, Akihiko Yano[3]☯, Atsushi Miyauchi[2], Masafumi Kawamura[2], Hideki Ito[3]

1 Department of Clinical Epidemiology, Kochi Medical School, Kochi University, Nankoku, Japan, 2 Department of Internal Medicine, Kochi Prefectural Hata-Kenmin Hospital, Sukumo, Japan, 3 Department of General Medicine, Kochi Health Sciences Center, Kochi, Japan, 4 Department of Healthcare Epidemiology, School of Public Health in the Graduate School of Medicine, Kyoto University, Kyoto, Japan

☯ These authors contributed equally to this work.
* sadak@kochi-u.ac.jp

**Data Availability Statement:** Data cannot be shared publicly because the data contain potentially identifying or sensitive patient information.

## Abstract

### Background

Coronavirus disease 2019 (COVID-19) patients reportedly have high bicarbonate concentration. However, its relationship to the disease progression are obscure.

### Methods

In this two-center retrospective study, we included COVID-19 patients with moderate severity between March 2020 and May 2021. We classified patients into three groups according to bicarbonate concentrations: high (>27 mEq/L), normal (21 to 27 mEq/L), and low (<21 mEq/L). The primary outcome was the time to clinical worsening defined by the requirement of intubation or death during 90 days. We evaluated high or low bicarbonate concentration during the clinical course related to the primary outcome using multivariable Cox proportional hazard models.

### Results

Of the 60 participants (median age 72 years), 60% were men. Participants were classified into high (13 patients), normal (30 patients), and low (17 patients) groups. Clinical worsening occurred in 54% of patients in the high group, 23% in the normal group, and 65% in the low group. Both high and low groups were associated with a higher clinical worsening rate: HR, 3.02 (95% CI, 1.05 to 8.63) in the high group; 3.49 (95% CI: 1.33 to 9.12) in the low group.

### Conclusion

Monitoring of bicarbonate concentrations may be useful to predict the prognosis.

However, data are available from the Ethics Committee of Kochi Medical School (contact via is21@kochi-u.ac.jp) for researchers who meet the criteria for access to confidential data.

**Funding:** The authors received no specific funding for this work.

**Competing interests:** The authors have declared that no competing interests exist.

## Introduction

Since December 2019, coronavirus disease 2019 (COVID-19) caused by SARS- CoV-2 has rapidly spread throughout the world and is still life-threatening despite the hard work of researchers.

High bicarbonate concentration was reportedly noted in COVID-19 patients with metabolic alkalosis [1]. Other studies have reported that hypokalemia is common among patients with COVID-19 [2, 3]. High bicarbonate concentration, metabolic alkalosis, and hypokalemia in COVID-19 patients are assumed to be due to the activation of the renin-angiotensin-aldosterone (RAA) system via the downregulation of angiotensin-converting enzyme 2 (ACE2) by SARS-CoV-2 [4]. There is a difference in the reports of two observational studies in terms of the prevalence of metabolic alkalosis and hypokalemia [1, 2]. Because patients with severe COVID-19 often exhibit metabolic acidosis and hyperkalemia resulting from multiple organ damage, the prevalence of high bicarbonate concentration, metabolic alkalosis, and hypokalemia might change according to COVID-19 severity.

The association between high bicarbonate concentration and disease progression is not fully elucidated. One report showed no difference in mortality between patients with and without metabolic alkalosis [1]. However, metabolic acidosis and low bicarbonate concentration are caused by multiple organ failure and related to high mortality in severe COVID-19 patients, so patients with metabolic acidosis and low bicarbonate concentration should be evaluated separately [5–8]. In addition, one case report showed that metabolic alkalosis developed with the worsening of COVID-19 severity in a patient. Thus, it is difficult to cross-sectionally evaluate the association between metabolic alkalosis and mortality without considering patients' clinical courses. One report evaluated the temporal change in bicarbonate concentration [9]. No difference in bicarbonate concentration at baseline was found between survivors and non-survivors, and a lower bicarbonate concentration was found at the last visit in non-survivors. Because the sequential time course of bicarbonate concentration was not evaluated and the disease severity at baseline was not taken into consideration in that study, the importance of a high bicarbonate concentration remains unclear.

The aim of our study was to investigate the sequential time course of bicarbonate levels with respect to changes in disease severity in COVID-19 patients with moderate disease severity.

## Materials and methods

### Study design and setting

We conducted a two-center retrospective cohort study at Kochi Health Sciences Center and Kochi Prefectural Hata-Kenmin Hospital. This study was performed following the Strengthening the Reporting of Observational studies in Epidemiology (STROBE) guidelines for reporting [10].

### Study population

We included patients with COVID-19 infection who were admitted to Kochi Health Sciences Center and Kochi Prefectural Hata-Kenmin Hospital from March 1, 2020, to May 31, 2021. All patients were over the age of 18 years and had SARS-CoV-2 infection confirmed by either reverse transcriptase-polymerase chain reaction or antigen test on respiratory tract samples and were started on oxygen therapy (moderate severity). The day when oxygen therapy was started was considered as the first day of inclusion. Patients were excluded from the study if

any of the following were applicable on the day of inclusion: the need for maintenance dialysis due to end-stage kidney disease; blood gases were not obtained; intubated at other hospitals.

## Data collection

From a review of electronic health records, we collected data such as age, sex, body mass index (BMI), comorbidity, medication history, vital sign, laboratory test, treatment limitations (limitations in providing life-sustaining therapies such as mechanical ventilation, cardiopulmonary resuscitation, and extracorporeal membranous oxygenation), and sequential organ failure assessment (SOFA) score. We collected serum pH, bicarbonate, and potassium concentrations that were obtained from venous or arterial blood gas analysis. Recent studies have suggested that the relationship between pH and $PaCO_2$ concentration obtained with venous blood gas and arterial blood gas sampling could allow venous blood to be used instead of arterial blood in analyses [11]. The $PaCO_2$ concentrations were collected only from arterial blood gas data. All blood gas analyses were performed using either an "ABL800 FLEX" or "ABL90FLEX" (Radiometer Medical ApS, Copenhagen, Denmark). Venous or arterial pH and $PaCO_2$ concentration were measured using selective electrodes. Bicarbonate concentration was calculated from pH and $PaCO_2$ using the Henderson-Hasselbalch equation. Potassium concentration was determined with direct potentiometry using ion-selective electrodes.

## Bicarbonate concentration categories

The patients were classified into three categories according to the bicarbonate concentration based on previous literature: high (>27 mEq/L), normal (21 to 27 mEq/L), and low (<21 mEq/L) [12–15]. Patients with bicarbonate concentration exceeding 27 mEq/L at least once within 7 days after inclusion were assigned to the high bicarbonate group while patients with bicarbonate concentration that decreased below 21 mEq/L at least once were assigned to the low bicarbonate group.

## Outcome measurement

The primary outcome was time to clinical worsening defined by the requirement of intubation or death within 90 days. Secondary outcomes included time to intubation, time to death, and time to clinical worsening within 28 days. Observations were censored if the event of interest did not occur, i.e., the patient was discharged from the hospital or did not visit the outpatient clinic after discharge.

## Statistical methods

Patient characteristics were described as the mean (standard deviation [SD]) or median (interquartile range [IQR]) as appropriate. Survival curves were plotted using the Kaplan-Meier method to compare the event probability at different points of time and to compare the three bicarbonate concentration groups. A log-rank test was applied to find the statistical significance among the three groups. As the primary analysis, Cox proportional hazard models were used to estimate hazard ratios (HRs) and 95% confidence intervals (CIs) for the association between bicarbonate concentration categories and clinical worsening within 90 days; the incidence of clinical worsening in the normal bicarbonate group was compared with those in the high bicarbonate and low bicarbonate groups. Multivariable analysis was performed after adjustment for treatment limitation.

We applied the same analyses as for the primary outcome to assess the association among three bicarbonate concentration groups and the following secondary outcomes: intubation within 90 days, 90-day mortality, and clinical worsening within 28 days.

To perform sensitivity analyses, we tested several Cox proportional hazard models to assess the robustness of the primary analysis. First, we adjusted the following covariates in the four models: age, sex, treatment limitation, and respiratory sub-SOFA score in model 1; age, sex, treatment limitation, respiratory sub-SOFA score, and renal sub-SOFA score in model 2; age, sex, treatment limitation, and $PaO_2/FiO_2$ ratio in model 3; age, sex, treatment limitation, respiratory sub-SOFA score, chronic pulmonary disease, hypertension, diabetes, and BMI in model 4. Second, we used the Cox regression model using penalized spline to evaluate the association between bicarbonate concentration as continuous value and clinical worsening within 90 days. GAM is an extension of the generalized linear model, where the predictors are related to the outcome via a smooth, possibly non-linear, function [16]. We adjusted with treatment limitation with a complete case analysis.

A p-value <0.05 was considered to indicate statistical significance. The analyses were performed using R software, version 4.0.3 (The R Foundation for Statistical Computing, Vienna, Austria. URL https://www.R-project.org/).

## Compliance with ethical standards

This study was conducted in accordance with the Declaration of Helsinki and the Ethical Guidelines for Medical and Health Research Involving Human Subjects in Japan. This study was approved by the Ethics Committee of Kochi Medical School (2021–007), and the Ethics Committee of Kochi Prefectural Hata-Kenmin Hospital. Patient data were anonymized and de-identified before the analysis. According to the Ethical Guidelines for Medical and Health Research Involving Human Subjects in Japan, the need to obtain written informed consent was waived due to the retrospective nature of the study.

## Results

### Patient characteristics

From March 1, 2020, to May 31, 2021, 112 patients with COVID-19 were admitted to the hospitals and 70 patients met the inclusion criteria. Of these, 10 patients were excluded. Finally, 60 patients were included in the analysis (Fig 1).

The median age of the included patients was 72 (IQR [64, 78]) years and 60% were males. The median BMI was 25.0 kg/m$^2$ (IQR [22.3, 28.7]). All patients were Japanese and 57% of patients had hypertension. The patients were classified into high bicarbonate (13 patients), normal bicarbonate (30 patients), and low bicarbonate (17 patients) groups according to bicarbonate concentrations. The median day from disease onset to inclusion was 8 (IQR [6, 9]) days. The median total SOFA score was 3 (IQR [2, 4]) and the median respiratory sub-SOFA score was 2 (IQR [2, 2]). Other patient characteristics at inclusion were summarized in Table 1.

### Blood gas data

The trend of pH, bicarbonate, $PaCO_2$, and potassium concentrations among the three bicarbonate concentration groups were shown in Fig 2. There was a tendency for the pH to decrease in the high bicarbonate group after day 3, but there was no significant change in the pH trend among the three groups (Fig 2a). The mean (±SD) bicarbonate concentration on days 1, 3, and 7 were 25.1 ± 3.0 mEq/L, 29.0 ± 5.2, and 33.5 ± 4.5 in the high bicarbonate group while

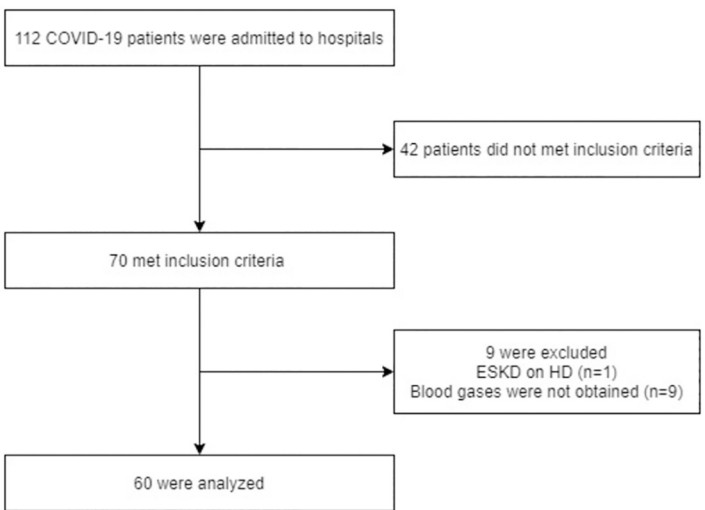

**Fig 1. Flow diagram.** ESKD, end-stage kidney disease; HD, hemodialysis.

19.4 ± 3.8 mEq/L, 21.1 ± 1.6 mEq/L, and 21.9 ± 3.4 mEq/L in the low bicarbonate group (Fig 2b). $PaCO_2$ tended to increase in the high bicarbonate group (Fig 2c). After day 3, there was a slight but not significant tendency for potassium to increase in the low bicarbonate group (Fig 2d).

## Association between bicarbonate concentration and the primary outcome

Clinical worsening occurred in 54% (7/13) of patients in the high bicarbonate group, 23% (7/30) in the normal bicarbonate group, and 65% (11/17) in the low bicarbonate group, respectively (Table 2). Fig 3 depicts the time to clinical worsening among the three bicarbonate concentration groups. The median times to clinical worsening among the three groups were significantly different; 11 (IQR [6, 41]) days in the high bicarbonate group, 16 (IQR [11, 25]) days in the normal bicarbonate, and 11 (IQR [7, 13]) days in the low bicarbonate, respectively; log-rank test, $p = 0.012$). The incidence of death and mechanical ventilation was higher in the high and low bicarbonate groups compared to the normal bicarbonate group (Table 2).

In the unadjusted analysis, high bicarbonate concentration and low bicarbonate concentration were associated with increased clinical worsening within 90 days (unadjusted HR 2.98 [95% CI 1.04 to 8.53], $p = 0.04$ in the high bicarbonate group, unadjusted HR: 3.80 [95% CI 1.46 to 9.89], $p = 0.006$ in the low bicarbonate group). After adjusting for treatment limitation, this association remained (adjusted HR: 3.02 [95% CI 1.05 to 8.63], $p = 0.04$ in the high bicarbonate group; adjusted HR: 3.49 [95% CI 1.33 to 9.12], $p = 0.01$ in the low bicarbonate group; Table 2).

## Secondary outcomes

The high and low bicarbonate groups had higher intubation within 90 days compared to the normal group (Table 2). For death within 90 days, the low bicarbonate group was higher than the normal group. When comparing the high bicarbonate group with the normal bicarbonate group, there was a trend toward more deaths in the high bicarbonate group, but the difference was not significant (Table 2). Compared with the normal bicarbonate group, clinical worsening within 28 days was significantly higher in the high bicarbonate group and the low bicarbonate group (Table 2).

**Table 1. Patient characteristics.**

| Characteristic | Overall | High bicarbonate | Normal bicarbonate | Low bicarbonate |
|---|---|---|---|---|
| | n = 60 | n = 13 | n = 30 | n = 17 |
| Age, years | 72 (64, 78) | 74 (71, 83) | 72 (65, 78) | 70 (61, 78) |
| Male, n (%) | 36 (60%) | 5 (38%) | 19 (63%) | 12 (71%) |
| BMI | 25.0 (22.3, 28.7) | 25.1 (22.4, 25.7) | 25.3 (23.4, 28.9) | 22.3 (20.9, 29.5) |
| Treatment limitation | 16 (27%) | 3 (23%) | 7 (23%) | 6 (35%) |
| Hypertension, n (%) | 34 (57%) | 6 (46%) | 17 (57%) | 11 (65%) |
| Diabetes, n (%) | 17 (28%) | 4 (31%) | 8 (27%) | 5 (29%) |
| Heart disease, n (%) | 9 (15%) | 1 (7.7%) | 3 (10%) | 5 (29%) |
| Dementia, n (%) | 4 (6.7%) | 2 (15%) | 0 (0%) | 2 (12%) |
| Chronic lung disease, n (%) | 4 (6.7%) | 0 (0%) | 3 (10%) | 1 (5.9%) |
| Body temperature, ˚C | 38.1 (37.4, 38.5) | 38.1 (37.9, 38.6) | 37.7 (37.0, 38.4) | 38.3 (38.2, 38.5) |
| Respiratory rate, bpm/min | 24 (22, 31) | 22 (20, 24) | 24 (22, 28) | 30 (25, 34) |
| SpO2, % | 92 (88, 94) | 91 (88, 93) | 92 (89, 94) | 93 (91, 94) |
| FiO2 | 0.32 (0.28, 0.50) | 0.28 (0.28, 0.32) | 0.30 (0.28, 0.50) | 0.36 (0.28, 0.60) |
| SOFA score | 3 (2, 4) | 3 (2, 3) | 2 (2, 3) | 4 (2, 6) |
| Respiratory sub-SOFA score | 2 (2, 2) | 2 (2, 2) | 2 (2, 2) | 2 (2, 2) |
| Renal sub-SOFA score | 0 (0, 0) | 0 (0, 0) | 0 (0, 0) | 0 (0, 1) |
| C-reactive protein, mg/L | 7.0 (4.4, 12.5) | 7.9 (2.7, 13.2) | 7.5 (3.4, 14.0) | 6.6 (4.8, 8.0) |
| Ferritin, ng/mL | 570 (310, 934) | 369 (185, 410) | 699 (365, 969) | 535 (261, 958) |
| Creatinine, mg/dL | 0.8 (0.7, 1.1) | 0.7 (0.6, 1.0) | 0.8 (0.7, 0.9) | 1.1 (1.0, 1.6) |
| Bicarbonate, mEq/L | 22.5 (21.6, 24.5) | 24.7 (23.0, 27.4) | 23.1 (22.1, 24.3) | 20.4 (17.8, 22.0) |
| Potassium, mEq/L | 3.8 (3.3, 3.9) | 3.5 (3.3, 3.8) | 3.8 (3.4, 3.9) | 3.8 (3.4, 3.9) |
| Medications | | | | |
| ACE inhibitors, n (%) | 3 (5.0%) | 1 (7.7%) | 1 (3.3%) | 1 (5.9%) |
| ARB, n (%) | 16 (27%) | 3 (23%) | 6 (20%) | 7 (41%) |
| Aldosterone blockers, n (%) | 2 (3.3%) | 1 (7.7%) | 0 (0%) | 1 (5.9%) |
| Renin blockers, n (%) | 0 (0%) | 0 (0%) | 0 (0%) | 0 (0%) |
| Loop diuretics, n (%) | 7 (12%) | 2 (15%) | 2 (6.7%) | 3 (18%) |

Statistics presented: Median (IQR) or n (%)

BMI; Body Mass Index, SpO2; peripheral oxygen saturation, FiO2; fraction of inspiratory oxygen, SOFA; peripheral oxygen saturation, ACE; angiotensin-converting enzyme, ARB; angiotensin receptor antagonist.

## Sensitivity analysis

The association between the bicarbonate categories and clinical worsening within 90 days remained similar in various multivariable Cox regression analyses (Table 3). We also performed a sensitivity analysis using continuous serum bicarbonate concentration as the exposure. Multivariable-Cox regression model using penalized spline, with the same adjustments as in the primary model, showed that both high and low bicarbonate concentrations were associated with an increased risk of clinical worsening within 90 days (Fig 4).

## Discussion

In our study, we evaluated the time course of bicarbonate concentrations in COVID-19 patients with moderate severity. High bicarbonate concentrations were noted in 19% of the included patients during the 7 days after inclusion. Of the patients who experienced high bicarbonate concentrations, 54% developed severe conditions.

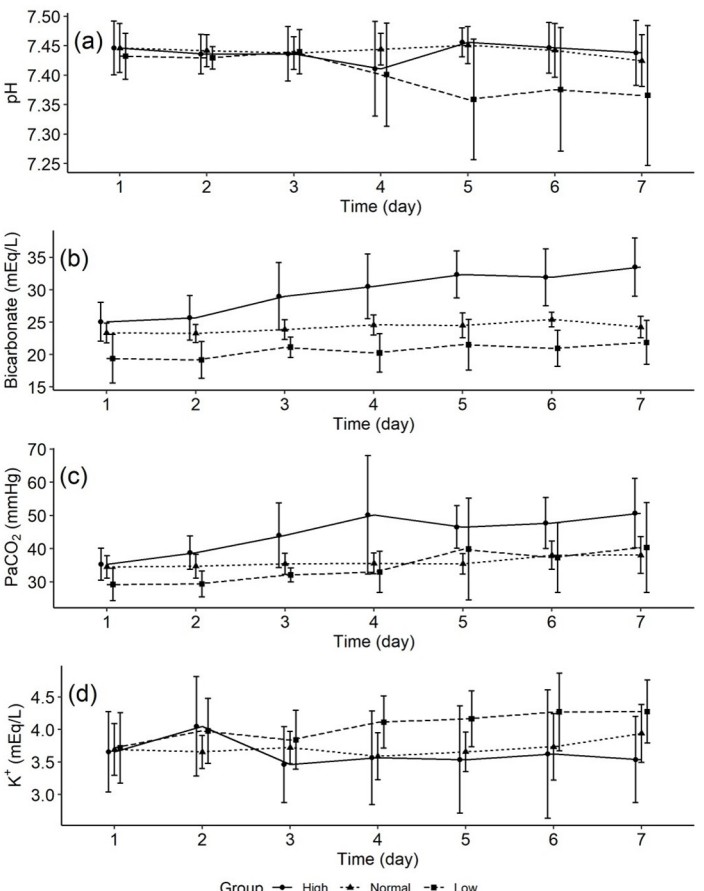

**Fig 2. pH, bicarbonate, PaCO$_2$, and potassium concentrations.**

Bicarbonate concentrations in blood might increase before the development of severe disease status. A recent report showed metabolic alkalosis and high bicarbonate levels were common among COVID-19 patients [1]. It is speculated that the downregulation of ACE2 by the activation of the RAA system leads to high bicarbonate concentrations and metabolic alkalosis. However, one other report found no increase in bicarbonate concentrations in COVID-19 patients [9]. One possible reason for this discrepancy is the disease severity of the enrolled patients. Although the former report did not show the outcomes in detail, the latter report included 23% of non-survivors. The patients with severe COVID-19 often exhibited metabolic acidosis and low bicarbonate concentrations caused by multiple organ damage, which is probably the reason why the bicarbonate concentrations were not high in the latter study. There was also a big difference in the prevalence of hypokalemia between the two aforementioned studies. One study reported hypokalemia in only 9% of the patients, [1] whereas 55% of the patients in the other study had hypokalemia [2]. These differences might be related not only to disease severity but also to disease time courses. In our study, potassium concentrations gradually decreased in the high bicarbonate concentration group up to day 5. Therefore, the prevalence of hypokalemia may change according to the timing of the evaluation.

The patients with high bicarbonate concentrations showed worse prognosis compared to those with low bicarbonate concentrations. There are two possible reasons why elevated bicarbonate is associated with poor prognosis. First, elevated bicarbonate levels may be an early

**Table 2. The association between bicarbonate concentration and outcomes.**

|  | Incidence | Crude HR (95% CI) | *p*-value | Adjusted* HR (95% CI) | *p*-value |
|---|---|---|---|---|---|
| **Primary outcome** | | | | | |
| Clinical worsening within 90 days, n (%) † | | | | | |
| High | 7/13 (54%) | 2.98 (1.04 to 8.53) | 0.042 | 3.02 (1.06 to 8.64) | 0.04 |
| Normal | 7/30 (23%) | ref | | ref | |
| Low | 11/17 (65%) | 3.80 (1.46 to 9.89) | 0.006 | 3.49 (1.33 to 9.12) | 0.01 |
| **Secondary outcomes** | | | | | |
| Intubation within 90 days, n (%) | | | | | |
| High | 6/13 (46%) | 6.10 (1.52 to 24.4) | 0.011 | 6.25 (1.56 to 25.0) | 0.01 |
| Normal | 3/30 (23%) | ref | | ref | |
| Low | 6/17 (35%) | 4.47 (1.11 to 18.0) | 0.035 | 4.91 (1.22 to 19.8) | 0.03 |
| Death within 90 days, n (%) | | | | | |
| High | 4/13 (54%) | 2.37 (0.63 to 8.89) | 0.200 | 2.54 (0.68 to 9.49) | 0.165 |
| Normal | 5/30 (17%) | ref | | ref | |
| Low | 8/17 (47%) | 4.01 (1.29 to 12.4) | 0.016 | 3.38 (1.08 to 10.5) | 0.035 |
| Clinical worsening within 28 days, n (%) | | | | | |
| High | 7/13 (54%) | 3.09 (1.08 to 8.84) | 0.035 | 3.09 (1.08 to 8.81) | 0.035 |
| Normal | 7/30 (23%) | ref | | ref | |
| Low | 10/17 (59%) | 3.54 (1.34 to 9.38) | 0.011 | 3.29 (1.24 to 8.74) | 0.017 |

HR, hazard ratio; CI, confidence interval;

*Adjusted for treatment limitation.

†Clinical worsening: Intubation or death for 90 days.

predictor of respiratory acidosis. Increased $PaCO_2$ and respiratory acidosis contribute to the severity of COVID-19. The $PaCO_2$ increased on Day 4 in the high bicarbonate group but only one patient met the definition of respiratory acidosis (pH<7.38 and $PaCO_2$>42mmHg). Higher bicarbonate may predict a subsequent rise in $PaCO_2$ and respiratory acidosis. Another reason is that elevated bicarbonate levels may reflect overactivity of the RAA system in patients with worsening COVID-19. Our results showed that $PaCO_2$ levels compensatorily increased without lowering pH in the high bicarbonate concentration group, which may indicate that

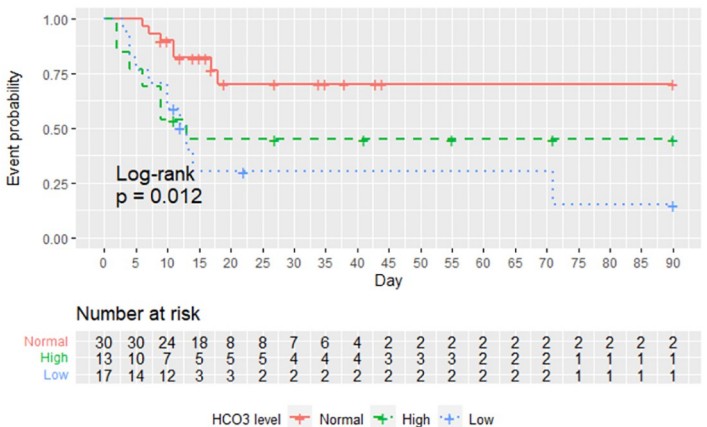

**Fig 3. Kaplan-Meier plot for time to clinical worsening within 90 days.** High bicarbonate (>27 mEq/L); normal bicarbonate (21 to 27 mEq/L); low bicarbonate (<21 mEq/L).

**Table 3. Sensitivity analysis for the association between bicarbonate concentration and clinical worsening within 90 days.**

| Models | High bicarbonate | | Low bicarbonate | |
|---|---|---|---|---|
| | Adjusted HR (95% CI) | *p*-value | Adjusted HR (95% CI) | *p*-value |
| Primary model | 3.02 (1.05 to 8.63) | 0.03 | 3.48 (1.33 to 9.12) | 0.01 |
| Model 1 | 3.27 (1.04 to 10.2) | 0.041 | 4.02 (1.49 to 10.8) | 0.006 |
| Model 2 | 3.24 (1.03 to 10.1) | 0.043 | 3.67 (1.33 to 10.1) | 0.012 |
| Model 3 | 3.22 (1.03 to 10.1) | 0.044 | 3.38 (1.26 to 9.05) | 0.016 |
| Model 4 | 3.29 (1.01 to 10.7) | 0.048 | 4.07 (1.49 to 11.1) | 0.006 |

HR, hazard ratio; CI, confidence interval, Reference is normal bicarbonate group

The primary model adjusted for treatment limitation,

Model 1 for age, sex, treatment limitation, and respiratory sub- sequential organ failure assessment (SOFA) score;

Model 2 for age, sex, treatment limitation, respiratory sub-SOFA score, and renal sub-SOFA score;

Model 3 for age, sex, treatment limitation, and PaO2/FiO2 ratio;

Model 4 for age, sex, treatment limitation, respiratory sub-SOFA score, chronic pulmonary disease, hypertension, diabetes, and body mass index.

high bicarbonate concentrations were related to metabolic alkalosis. The previous study [1] found no difference in mortality between COVID-19 patients with and without metabolic alkalosis, but the study did not separate the patients with metabolic acidosis. Metabolic acidosis is a well-known risk factor for mortality in COVID-19 patients [5]. In our study, we could confirm the worst prognosis in the patients with low bicarbonate concentrations, which indicated that the patients exhibited metabolic acidosis. In the course of the activation of the RAA system, angiotensin II, via activation of angiotensin type 1a receptor, reportedly promotes inflammatory responses in COVID-19 patients [17]. Therefore, the high bicarbonate concentrations noted in our study population possibly reflects the hyperactivation of the RAA system in patients with worsening COVID-19.

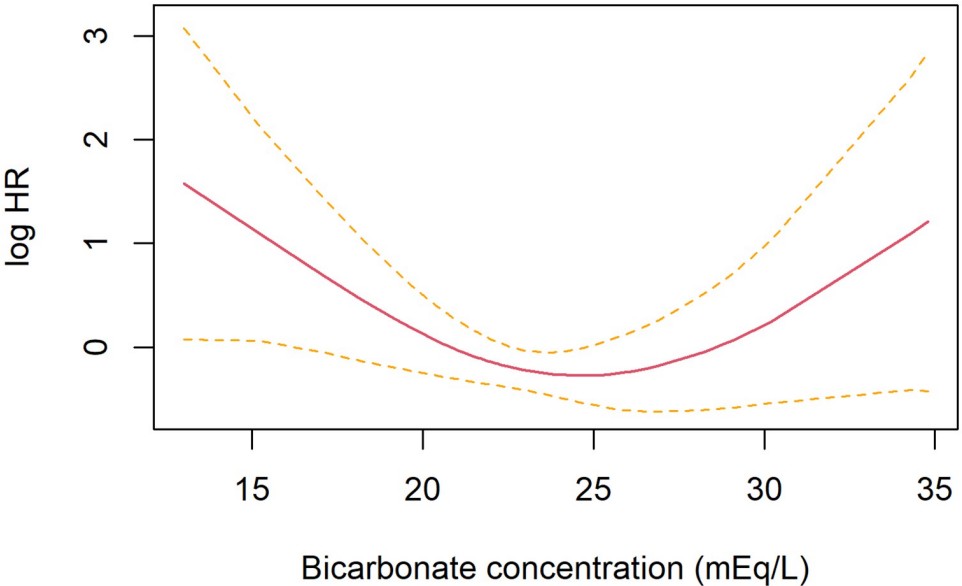

**Fig 4. The Cox regression model using penalized spline results for relationships between bicarbonate concentration and clinical worsening within 90 days.** The black solid line indicates log hazard ratio (HR) and the dotted lines indicate standard error (SE).

Our patient had a slightly higher pH on Day 1 (Fig 2a), but whether this was due to disease specificity is not known because it was not compared to non-COVID-19 patients in our study. Although evidence is limited, higher pH has also been reported in patients with H1N1 influenza [18, 19]. Even before hypotension occurs, fever and sepsis can lead to an increase in pH via hyperventilation [20]. Therefore, the high pH on Day 1 may not be attributed to the disease specificity of COVID-19.

There are some limitations to our study. First, bicarbonate concentrations were evaluated using both venous and arterial blood analysis. The utility of venous blood gas should be evaluated further although it is a less invasive procedure than arterial blood gas analysis. However, previous meta-analysis studies indicated that venous and arterial bicarbonate levels are reasonably close [11, 21], so we believe that venous blood gas analysis could also be useful for bicarbonate monitoring in COVID-19 patients. Second, the bicarbonate concentration obtained at baseline may not be a useful predictor of prognosis. Since a few patients showed high bicarbonate concentrations at baseline, we cannot exclude the requirement for sequential monitoring of patients with normal bicarbonate concentrations at baseline. The third was the use of a Cox model adjusted for six covariates for fewer outcomes. However, the standard errors of the regression coefficients were small, and we believe that we have achieved a stable estimation.

## Conclusions

High bicarbonate concentrations during the clinical course in COVID-19 patients with moderate disease status were related to a worse prognosis. Sequential monitoring of bicarbonate concentrations may be useful to predict the prognosis of COVID-19 patients.

## Acknowledgments

The authors would like to thank Mrs. Kimiko Shirotake for her assistance in data management and all medical staff members who cooperated in data collection.

## Author Contributions

**Conceptualization:** Ken-ei Sada, Ryohei Yamamoto, Akihiko Yano.

**Data curation:** Ryohei Yamamoto, Akihiko Yano, Atsushi Miyauchi, Masafumi Kawamura, Hideki Ito.

**Formal analysis:** Ryohei Yamamoto.

**Investigation:** Ken-ei Sada, Ryohei Yamamoto, Akihiko Yano.

**Methodology:** Ken-ei Sada, Ryohei Yamamoto, Akihiko Yano.

**Project administration:** Ken-ei Sada.

**Supervision:** Masafumi Kawamura, Hideki Ito.

**Visualization:** Ryohei Yamamoto.

**Writing – original draft:** Ken-ei Sada, Ryohei Yamamoto, Akihiko Yano.

**Writing – review & editing:** Ken-ei Sada, Ryohei Yamamoto, Akihiko Yano, Atsushi Miyauchi, Masafumi Kawamura, Hideki Ito.

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
