## [Decision Letter · Decision Letter 0]

28 Mar 2022

PONE-D-22-03671Bicarbonate concentration as a predictor of prognosis in moderately severe COVID-19 patients: a multicenter retrospective studyPLOS ONE

Dear Dr. Sada,

Thank you for submitting your manuscript to PLOS ONE. After careful consideration, we feel that it has merit but does not fully meet PLOS ONE’s publication criteria as it currently stands. Therefore, we invite you to submit a revised version of the manuscript that addresses the points raised during the review proces。It is great interests to find valuable biomarkers for COVID19 infection and its prognosis.  The current article will draw high interests from the readers.Before accepting the article, two experts raised several concerns, specially in relation with respiratory factors affecting bicarbonate concentration. Please submit your revised manuscript by May 12 2022 11:59PM. If you will need more time than this to complete your revisions, please reply to this message or contact the journal office at plosone@plos.org. Please include the following items when submitting your revised manuscript:A rebuttal letter that responds to each point raised by the academic editor and reviewer(s). You should upload this letter as a separate file labeled 'Response to Reviewers'.A marked-up copy of your manuscript that highlights changes made to the original version. You should upload this as a separate file labeled 'Revised Manuscript with Track Changes'.An unmarked version of your revised paper without tracked changes. You should upload this as a separate file labeled 'Manuscript'.

We look forward to receiving your revised manuscript.

Kind regards,

Tatsuo Shimosawa, M.D., Ph.D.

Academic Editor

PLOS ONE

Journal Requirements:

Reviewers' comments:

Reviewer's Responses to Questions

**Comments to the Author**

1. Is the manuscript technically sound, and do the data support the conclusions?

Reviewer #1: Partly

Reviewer #2: Yes

2. Has the statistical analysis been performed appropriately and rigorously? 

Reviewer #1: No

Reviewer #2: Yes

3. Have the authors made all data underlying the findings in their manuscript fully available?

Reviewer #1: Yes

Reviewer #2: Yes

4. Is the manuscript presented in an intelligible fashion and written in standard English?

Reviewer #1: Yes

Reviewer #2: Yes

5. Review Comments to the Author

Reviewer #1: The authors investigated the association between serum bicarbonate levels and clinical outcomes in patients with COVID-19.

Not only respiratory failure but also multiple organ failure causes the disorder of acid-base balance. Therefore, the interpretation of the disorder caused by COVID-19 is of great interest.

However, there are several concerns in this study.

First, the author did not show the rationale for the definitions of high, normal, and low bicarbonate. As this study was conducted retrospectively, the standards should be shown in justification.

Second, the authors conducted multivariate Cox regression analyses with several models. However, the numbers of patients who reached the outcomes were around ten in each group.

Consequently, I think that the authors adjusted too many parameters. Due to the small number of patients, I do not think that appropriate multivariable Cox models were established in this study.

Third, the authors should explain “treatment limitation” in detail. There is no information in this manuscript.

Fourth, as the patients were not intubated, the authors should show how they calculated FiO2.

Reviewer #2: The authors have investigated the utility of measuring HCO3 concentrations to predict prognosis in COVID-19 patients. The manuscript contains interesting findings that may be of interest to readers. However, the points discussed below need to be addressed.

Major Points

1) The COVID-19 patients in this study had a higher pH on Day 1. Consider whether this should be attributed to disease specificity, and describe the differences in the pH results between non-COVID-19-infected and COVID-19-infected patients.

2) Table 2 shows that patients with COVID-19 have higher pH from Day 1 and that respiratory alkalosis is suspected due to the low PaCO2 values. The authors should evaluate the patients for alkalosis on Day 1.

3) In the discussion, the authors describe metabolic alkalosis as an important change in severe disease (lines 246–249). According to the graph of pH changes shown in Table 2, the pH decreased on Day 4 in the high HCO3 group. An abnormal PaCO2 increase may suggest an improper functioning of respiratory compensation, and the presence of respiratory acidosis may also be detected. Careful consideration should be given to the possibility that PaCO2 increases and respiratory acidosis might contribute to the severity of COVID-19.

4) Diuretics are an important cause of metabolic alkalosis and hypokalemia. The history of diuretic use should be described.

6. PLOS authors have the option to publish the peer review history of their article (what does this mean?). If published, this will include your full peer review and any attached files.

Reviewer #1: No

Reviewer #2: No

---

## [Author Response · Author response to Decision Letter 0]

26 Apr 2022

We have attached our reply as the file.

---

## [Decision Letter · Decision Letter 1]

23 May 2022

PONE-D-22-03671R1Bicarbonate concentration as a predictor of prognosis in moderately severe COVID-19 patients: a multicenter retrospective studyPLOS ONE

Dear Dr. Sada,

Thank you for submitting your manuscript to PLOS ONE. After careful consideration, we feel that it has merit but does not fully meet PLOS ONE’s publication criteria as it currently stands. Therefore, we invite you to submit a revised version of the manuscript that addresses the points raised during the review process.

Please reply to the concern on your statistical analysis to the reviewer.

We look forward to receiving your revised manuscript.

Kind regards,

Tatsuo Shimosawa, M.D., Ph.D.

Academic Editor

PLOS ONE

Journal Requirements:

Reviewers' comments:

Reviewer's Responses to Questions

**Comments to the Author**

1. If the authors have adequately addressed your comments raised in a previous round of review and you feel that this manuscript is now acceptable for publication, you may indicate that here to bypass the “Comments to the Author” section, enter your conflict of interest statement in the “Confidential to Editor” section, and submit your "Accept" recommendation.

Reviewer #1: All comments have been addressed

Reviewer #2: All comments have been addressed

2. Is the manuscript technically sound, and do the data support the conclusions?

Reviewer #1: Yes

Reviewer #2: Yes

3. Has the statistical analysis been performed appropriately and rigorously? 

Reviewer #1: No

Reviewer #2: Yes

4. Have the authors made all data underlying the findings in their manuscript fully available?

Reviewer #1: Yes

Reviewer #2: Yes

5. Is the manuscript presented in an intelligible fashion and written in standard English?

Reviewer #1: Yes

Reviewer #2: Yes

6. Review Comments to the Author

Reviewer #1: The authors replied to almost all the concerns I pointed out. Nonetheless, I have still concerned about too many fittings in the multivariable Cox regression model.

Generally, events per variable (EPV)>10 is recommended, and even one of the authors' suggested references evaluated EPV=3 (least) and EPV=5 (Stat Med. 2016 Mar 30;35(7):1159-77.) However, the EPV in the multivariable Cox regression model in this study was less than 2, which was extremely low.

In my opinion, the result shown in this study is plausible. However, I do not understand that the authors stuck to the multivariable model. I think univariable or only a few parameters fitting Cox regression models are enough to show the result.

Reviewer #2: First, I apologize for the confusion due to my previous comments on Figure 2 and Table 2.

I would like to thank the authors for responding to my initial comments.

I do not have further questions.

7. PLOS authors have the option to publish the peer review history of their article (what does this mean?). If published, this will include your full peer review and any attached files.

Reviewer #1: No

Reviewer #2: No

---

## [Author Response · Author response to Decision Letter 1]

3 Jun 2022

Reviewer #1: 

Reviewer comment

The authors replied to almost all the concerns I pointed out. Nonetheless, I have still concerned about too many fittings in the multivariable Cox regression model.

Generally, events per variable (EPV)>10 is recommended, and even one of the authors' suggested references evaluated EPV=3 (least) and EPV=5 (Stat Med. 2016 Mar 30;35(7):1159-77.) However, the EPV in the multivariable Cox regression model in this study was less than 2, which was extremely low.

In my opinion, the result shown in this study is plausible. However, I do not understand that the authors stuck to the multivariable model. I think univariable or only a few parameters fitting Cox regression models are enough to show the result.

Response

Thank you for the comment. We agree with your suggestion that univariable or only a few parameters fitting Cox regression models are enough to show the result. We revised our primary analysis to a model adjusted for only treatment limitations.

Since it would be an outcome reporting bias not to include the results of pre-planned analyses, we would like to remain the results of these sensitivity analyses.

The main results and sensitivity analysis tables and the corresponding text sections have been changed accordingly. 

Reviewer #2: 

First, I apologize for the confusion due to my previous comments on Figure 2 and Table 2.

I would like to thank the authors for responding to my initial comments.

I do not have further questions.

Response

Thank you.

---

## [Editor Report · Decision Letter 2]

6 Jun 2022

Bicarbonate concentration as a predictor of prognosis in moderately severe COVID-19 patients: a multicenter retrospective study

PONE-D-22-03671R2

Dear Dr. Sada,

We’re pleased to inform you that your manuscript has been judged scientifically suitable for publication and will be formally accepted for publication once it meets all outstanding technical requirements.

Kind regards,

Tatsuo Shimosawa, M.D., Ph.D.

Academic Editor

PLOS ONE
---

## [Editor Report · Acceptance letter]

14 Jun 2022

PONE-D-22-03671R2 

Bicarbonate concentration as a predictor of prognosis in moderately severe COVID-19 patients: a multicenter retrospective study 

Dear Dr. Sada:

I'm pleased to inform you that your manuscript has been deemed suitable for publication in PLOS ONE. Congratulations! Your manuscript is now with our production department. 

Kind regards, 

on behalf of

Prof. Tatsuo Shimosawa 

Academic Editor

PLOS ONE